# Analysis of the Risk Factors Associated with Hearing Loss of Infants Admitted to a Neonatal Intensive Care Unit: A 13-Year Experience in a University Hospital in Korea

**DOI:** 10.3390/ijerph17218082

**Published:** 2020-11-02

**Authors:** Kyu Young Choi, Bum Sang Lee, Hyo Geun Choi, Su-Kyoung Park

**Affiliations:** 1Department of Otorhinolaryngology-Head and Neck Surgery, Kangnam Sacred Heart Hospital, Hallym University College of Medicine, Seoul 07441, Korea; coolq0@hallym.or.kr (K.Y.C.); annoce@naver.com (B.S.L.); 2Department of Otorhinolaryngology-Head and Neck Surgery, Hallym Sacred Heart Hospital, Hallym University College of Medicine, Anyang 14068, Korea; hgchoi@hallym.or.kr

**Keywords:** evoked potentials, auditory, brain stem, hearing loss, infant, newborn, intensive care units, neonate, risk factors

## Abstract

Early detection of hearing loss in neonates is important for normal language development, especially for infants admitted to the neonatal intensive care unit (NICU) because the infants in NICU have a higher incidence of hearing loss than healthy infants. However, the risk factors of hearing loss in infants admitted to the NICU have not been fully acknowledged, especially in Korea, although they may vary according to the circumstances of each country and hospital. In this study, the risk factors of hearing loss in NICU infants were analyzed by using the newborn hearing screening (NHS) and the diagnostic auditory brainstem response (ABR) test results from a 13-year period. A retrospective chart review was performed using a list of NICU infants who had performed NHS from 2004 to 2017 (*n* = 2404) in a university hospital in Korea. For the hearing loss group, the hearing threshold was defined as 35 dB nHL or more in the ABR test performed in infants with a ‘refer’ result in the NHS. A four multiple number of infants who had passed the NHS test and matched the age and gender of the hearing loss group were taken as the control group. Various patient factors and treatment factors were taken as hearing loss related variables and were analyzed and compared. From the 2404 infants involved, the prevalence of hearing loss was 1.8% (*n* = 43). A comparison between the hearing loss group (*n* = 43) and the control group (*n* = 172) revealed that history of sepsis, peak total bilirubin, duration of vancomycin use, days of phototherapy, and exposure to loop-inhibiting diuretics were significantly different, and can be verified as significant risk factors for hearing loss in NICU infants.

## 1. Introduction

Hearing is essential for normal development of the child’s language [1]. Hearing impairment in the early stage of life may lead to cognition, learning, and social degradation [2]. Without appropriate treatment, hearing disturbance can lead not only to communication problems or academic failure but also to social and psychological difficulties [3]. Many studies have claimed that early identification of hearing loss and intervention contribute to better outcomes in language development over the entire lifetime [1,4,5]. The incidence of bilateral severe hearing loss in all newborns is 2 to 3 per 1000 [6]. However, the newborns in neonatal intensive care units (NICUs) have a much higher incidence of hearing loss than that of normal babies; 20 to 50 per 1000 newborns [7]. In 2000 and 2007, the guidelines of The Joint Committee on Infant Hearing (JCIH) and other guidelines recommended that all infants undergo newborn hearing screening (NHS) by automated auditory brainstem response (AABR) and otoacoustic emissions (OAEs) tests before 1 month of age, diagnosis for infants with the ‘refer’ result in NHS should be made before 3 months of age, and intervention should be initiated before 6 months of age; altogether, these guidelines are called the ‘1-3-6’ early hearing detection and intervention (EHDI) guidelines [2,8,9,10]. Several publications, including JCIH guidelines, report that the risk factors for hearing impairment in the newborns include NICU stay of more than 5 days, duration of assisted ventilation, low Apgar scores (a test to evaluate a newborn’s physical condition referring appearance, pulse, grimace, activity, and respiration), ototoxic drug exposure, craniofacial anomalies, in utero infection, sepsis, and meningitis [2,8,9,10,11,12,13,14].

However, the reported risk factors of hearing loss vary among reports, which may be explained with different conditions such as types of NICU, city- and country-specific circumstances among studies. Although, every country and every hospital should acknowledge their own risk factors for hearing loss, to the best of the author’s knowledge, there are few studies on the risk factors of hearing loss in NICU hospitalized infants in Asia, including Korea. Herein, we evaluated and analyzed the risk factors of hearing loss of NICU infants in Seoul, Korea, over a 13-year period.

## 2. Materials and Methods

### 2.1. Subjects and Definitions

Long-term retrospective data were collected for 2404 infants who had been hospitalized in NICU longer than 24 h and underwent NHS by means of AABR (MB11 BERAphone, MAICO diagnostics, Berlin, Germany) or both AABR and OAEs (Audioscreener, GSI, MN, USA) from July 2004 to November 2017 in Kangnam Sacred Heart Hospital, Seoul, Korea. NHS was performed immediately before discharge when the infant was in good condition. Sound stimulus intensity in AABR was set at 35 dB nHL. In the case of a ‘refer’ result for any ear in AABR, infants were referred for diagnostic ABR test (Bio-logic Navigator Pro, Natus Medical, Taastrup, Denmark) using an air conduction click stimulus at an otorhinolaryngology outpatient clinic. Hearing loss was defined as an elevated ABR response threshold (≥35 dB nHL) in either ear. All the NICU infants in the study period with the results of NHS were included. Exclusion criteria included those infants without any of NHS results, and those lacking medical information for any patient and treatment factors as described below, such as infants born in other hospitals and admitted to NICU in the study hospital.

### 2.2. Classification of Groups and Methods

The hearing loss group included babies whose hearing thresholds were ≥35 dB in ABR, which included unilateral hearing loss. For the control group, to increase statistical power, four times the numbers of infants in the hearing loss group (the first 172 consecutive cases) who had passed in the NHS test were taken [15], and then they were matched to the hearing loss group in terms of gestational weeks and gender. The birth year of the infants was within 3 years to minimize changes in care practices over the admission period.

For the patient factors, the following patient and treatment variables were obtained from the medical records of each group: gender, history of germinal matrix hemorrhage, respiratory distress syndrome, cerebral hemorrhage, sepsis, pneumonia, facial anomaly, chromosome anomaly, congenital metabolic disorder, intrauterine infection (toxoplasmosis, rubella, cytomegalovirus, herpes virus), Apgar score at 1 min and 5 min, peak total bilirubin, and peak direct bilirubin. For the treatment factors, number of days in NICU, duration of ventilation, days of phototherapy, ototoxic drug usage and duration (e.g., aminoglycoside, vancomycin, or loop-inhibiting diuretics), and usage of surfactant or blood transfusion were taken.

### 2.3. Statistical Analysis

Quantitative variables were described as the number of non-missing values, mean, and standard deviation. Qualitative variables were described as the number and percentage of infants. Missing values were not included in the calculation. For quantitative data where the standard deviation (SD) exceed mean values, we presented median and interquartile range (quartile 1, quartile 3) to avoid bias due to unnormal distribution of data, and in other cases, we expressed the mean and SD.

All statistical analyses were performed using SPSS 25 (IBM Co., Armonk, NY, USA). Initially, all data were analyzed using the Kolmogorov–Smirnov test to assess for normality. Group comparison for qualitative data was performed by the Pearson chi-square test, whereas the Student’s *t*-test was used for quantitative data. *p*-values < 0.05 were considered statistically significant.

### 2.4. Ethics Statement

This study was reviewed and approved by the Institutional Review Board (IRB) of the Ethics Committee of Kangnam Sacred Heart Hospital of the Hallym University College of Medicine, Korea (approval IRB No. HKS201809006). Informed consent was waived by the board due to the retrospective nature of this study. Patients’ confidentiality was protected and their information was anonymized before statistical analysis.

## 3. Results

### 3.1. Status of Hearing Screening and Hearing loss in NICU Infants

From 2004 to 2017, 2404 infants were admitted to the NICU nursery and underwent hearing screening tests. Among them, 2227 infants had ‘pass’ and 177 infants had ‘refer’ in the NHS. The average stay at the NICU was 20.3 ± 31.0 days. The overall referral rate was 7.4% (both ears: 3.2%; unilateral: 4.1%). Of the 177 referred infants, 60 (33.9%) underwent diagnostic ABR at an otorhinolaryngology outpatient clinic. The average age was 17.5 ± 27.9 days at the time of NHS (for the whole 2404 infants), and 193.2 ± 201.3 days (6.4 ± 6.7 months) at the time of diagnostic ABR (for the 60 infants). Among the 60 infants, 43 were finally diagnosed with sensorineural hearing loss (26 bilateral and 17 unilateral). The overall prevalence of hearing loss was 1.8% among all infants. Since not all of the referred infants underwent diagnostic ABR tests, the prevalence of hearing loss was corrected for the ABR rate, resulting in 5.3% (bilateral: 3.2%; unilateral: 2.1%). The 43 infants diagnosed with hearing impairment were classified as the hearing loss group, and 172 infants (4 multiples of the hearing loss group) who had passed NHS and matched age and gender to the hearing loss group were selected as the control group. The study process is depicted in Figure 1.

### 3.2. Risk Factors Associated with Hearing Loss in NICU Infants

The average ages when the NHS was performed were 44.5 (7, 77.5) days for the control group and 55.0 (13.0, 77.0) days for the hearing loss group, with no significant difference between the two groups. The average age when the ABR test was performed for the hearing loss group was 119.0 (64.0, 174.0) days (Table 1).

In the hearing loss group, the following patient risk factors were identified with high incidence: sepsis (62.8%), respiratory distress syndrome (41.9%), congenital metabolic abnormality (25.6%), and pneumonia (16.3%). The order of their prevalence was the same in the control group, but the incidence of the risk factors was lower than that of the hearing loss group. Among these factors, the presence of sepsis and peak total bilirubin was significantly higher for the hearing loss group compared to the control group (*p* = 0.003 and *p* = 0.001, respectively; Table 2).

For the treatment factors, the following factors were highly identified in the hearing loss group: transfusion (58.1%), vancomycin use (44.2%), surfactant use (41.9%), and loop-inhibiting diuretics use (39.5%). However, comparison between hearing loss group and control group showed a statistically significant difference for phototherapy days (*p* = 0.002), duration of vancomycin use (*p* = 0.024), and number of infants using loop-inhibiting diuretics (*p* = 0.003). The cumulative duration of aminoglycoside or ventilation days was longer in the hearing loss group than in the control group, however, there was no significant difference between the two groups. All other factors were not significantly different in the statistical analysis (Table 3).

## 4. Discussion

For the early diagnosis and intervention of hearing loss, the ‘1-3-6’ EHDI guidelines are generally recommended [2,8,10,12,16]. In the present study, the NHS was performed at an average age of 17.5 days, i.e., within 1 month of age, but the diagnostic ABR was performed at 6.4 months of age on average, i.e., beyond 3 months of age. The main reason for the delayed ABR tests can be the fact that the intensive care unit infants should be in a better condition in order to perform ABR (taking sedatives at the otolayngology outpatient clinics after discharge from the NICU) in comparison with NHS (much easier with no sedatives). In this study, the average age at the time of NHS was 44.5 (7.0, 77.5) days in the control group and 55.0 (13.0, 77.0) days in the hearing loss group. Both are beyond 1 month after birth; however, considering that most infants were preterm when they were born (average of 31.0 and 30.6 gestational weeks) for the control and hearing loss group, respectively), we estimated the corrected NHS days and found that they had NHS at 37.4 and 38.4 gestational weeks; both values are within the recommended 1-month period.

The average hospitalization period in the NICU was 52.0 (7.0, 84.8) days for the control group and 63.0 (14.0, 88.0) days for the hearing loss group. The age of the diagnostic ABR of the hearing loss group was 119.0 (64.0, 174.0) days. In this study, the follow-up ABR rate of all referred infants was 33.9%, which was lower than that of the Taiwan study (81.4%) [17]. Thus, efforts to improve the follow-up diagnostic ABR rate after NHS and, furthermore, to improve the diagnostic system are needed. In the present study, the prevalence of bilateral hearing loss was 3.2% and that of unilateral hearing loss was 2.1% (after correction for the follow-up ABR rate). In the Taiwan study, the corrected bilateral HL prevalence rate was 1.84% [17]. The prevalence rate for the Dutch infants was 1.7%; other studies reported the bilateral profound hearing loss rate in NICU infants as 0.8–2.0% (not corrected for ABR rate; these values could be up to 10 times those of healthy babies) [7,8,18,19].

The auditory system is known to be one of the most susceptible organs to toxic agents. Hyperbilirubinemia is a common cause of hearing impairment in newborns and can also have toxic effects on the central nervous system such as cerebral palsy, epilepsy, or cognitive deficits [20]. The treatments for neonatal hyperbilirubinemia are phototherapy and blood exchange transfusion. In general, hyperbilirubinemia requiring these treatments is determined by considering multiple factors, such as gestational weeks at birth, daily increase in bilirubin level, total serum bilirubin, and other laboratory test results [21]. In the JCIH guidelines and other studies, hyperbilirubinemia that requires exchange transfusion is a risk indicator associated with permanent congenital hearing loss [2,20,22]. In the present study, peak total bilirubin level and the duration of phototherapy, but not the number of infants who received transfusion, were significantly different between the hearing loss and control groups. Although severe neonatal hyperbilirubinemia with exchange transfusion is associated with hearing loss, not all transfused newborns with severe hyperbilirubinemia develop hearing deficits [22,23]. To our knowledge, there has been few published studies that would have investigated the duration of phototherapy as a factor causing congenital permanent hearing loss. The duration of phototherapy could be a risk factor for hearing loss because hyperbilirubinemia is more common and severe in preterm than in term infants and is usually accompanied by other serious diseases, and the duration of phototherapy indirectly reflects the severity of hyperbilirubinemia because it is performed as an initial treatment of severe hyperbilirubinemia before exchange transfusion [22]. Further studies on the relationship between hearing impairments and the duration of phototherapy and exchange transfusion are warranted to elucidate the underlying mechanisms of the increased risks.

Several studies have demonstrated that exposure to ototoxic medications such as aminoglycoside, vancomycin, and loop-inhibiting diuretics are risk factors of congenital or delayed hearing loss [2,13,23,24]. In this study, among these medications, only the duration of exposure to loop-inhibiting diuretics and vancomycin differed significantly between the hearing loss and control groups, while the fact of exposure to ototoxic drugs as such was not a significant risk indicator. In a 26,341-cohort study that evaluated the risk indicators for congenital and delayed-onset hearing loss in the US, a NICU stay of more than 5 days and exposure to loop-inhibiting diuretics were not associated with an increased risk of hearing loss, unlike the risk indicators of JCIH guidelines [23]. Ototoxic medications are already well known as hearing loss risk indicators, and most doctors are specifically monitoring the use of these medications, and this monitoring can be one of the reasons for no significant change (or just a small increase) in the number of newborns with hearing loss compared to those without these medications.

Several publications have evaluated the risk factors of hearing loss in NICU; a Dutch study found that dysmorphic features, low 1-min Apgar scores, sepsis, meningitis, cerebral bleeding, and cerebral infarction were associated with hearing loss [25], and a Mexican study conducted over 15 years reported that low birth weight, longer NICU stay, prevalence of blood exchange, intraventricular hemorrhage, higher serum bilirubin levels, and meningitis were the main risk indicators of hearing loss [8]. An interesting finding was reported by a study of Western Sicily; as the number of coexisting risk factors increased, the percentage value of hearing loss in infants and the degree of hearing loss increased [13]. Only a few studies had been performed and reported in the last 20 years to evaluate the risk factors for the NICU infants in Asia. In 2015, an Indian study consisting of 500 infants of term and preterm reported that NICU stay, low birth weight, and hypoxia were the risk factors for hearing loss [26]. A Japanese study in 2019 studied risk factors of hearing loss in 1071 NICU infants, and reported that oxygen administration and chromosomal aberrations were the risk factors [27]. However, a Malaysian study in 2020, which evaluated 2713 infants in NICU, reported that only craniofacial anomalies were the risk factors of hearing loss [28]. A Chinese survey of 616,940 households consisting of disabled people and children aged 0–6 years in 2020, reported that lower annual family income, male children, larger household size, single-mother family, and lower levels of maternal and paternal education were the social risk factors for hearing loss; however, this was just a survey and not a study of NICU infants [29]. The Belgian recommendation in 2015 was the only national study for risk factors of hearing loss, after the 2007 JCIH recommendation [30]. They reported congenital infections such as cytomegalovirus, toxoplasmosis, and syphilis, a family history of hearing loss, consanguinity in (grand)parents, malformation syndromes, and fetal alcohol syndrome as ‘high’ level quality evidence of neonatal risk factors for hearing loss, bilirubin toxicity and hyperbilirubinemia as ‘moderate’ level, very low birth weight, low Apgar score, and hospitalization in the NICU as ‘very low’ to ‘low’ levels, and ototoxic drugs as ‘very low’ level. The risk factors for hearing loss in NICU infants in Korea have not been evaluated and reported in the literature up to now. This study is the first one to evaluate hearing loss in infants in Korea with more than 2000 cases, and the risk factors of hearing loss with more than 200 cases. We should be aware that studies conducted in different countries have reported different risk factors for hearing loss, likely because of differences in geographic characteristics and critically ill NICU patients.

The limitations of this study for the analysis of the status of NHS were that it investigated only up to until the final diagnosis of hearing loss, and no follow-up was conducted after the intervention for hearing loss such as hearing aids or cochlear implantation because most rehabilitation treatments were performed at other children’s hospitals and no medical records were available to us. Another limitation was the small size of the hearing loss group (*n* = 43). Although, the statistical power of the study was increased by selecting four controls per case (the additional gain is known to be small if the case/control ratio exceeds 1:4) [17], segmentation by risk factors may result in a smaller size of the groups, which may not produce statistically significant results. Therefore, caution is needed in the interpretation of these results. For example, mechanical ventilation, family history of sensorineural hearing loss, and the use of aminoglycosides were significant factors associated with hearing loss in other studies but not in this study. The retrospective collection of the data for the risk factors in NICU infants was also a potential weakness of this study design. In addition, this study analyzed 13 years of NICU data, and there have been many medical advances in neonatal intensive care over this time period, and these advances may have affected the variables associated with hearing loss in NICU. Further prospective study is needed to evaluate whether similar associations between risk factors and HL are observed in different countries, patient populations, and health care settings. We think that every NICU should evaluate their own risk factors and undertake persistent monitoring to prevent hearing loss in their NICU infants with hearing loss risk indicators.

## 5. Conclusions

In summary, history of sepsis, peak total bilirubin, duration of vancomycin use, the number of days of phototherapy, and the use of LIDs were significant risk factors for hearing loss in NICU infants in Korea. These factors should be considered in NICU infant care and also in counseling parents of the NICU graduates.

## Figures and Tables

**Figure 1 ijerph-17-08082-f001:**
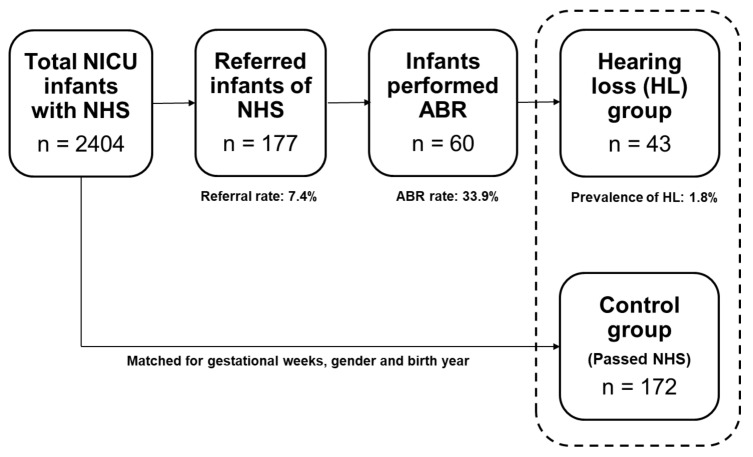
Study process and the results of newborn hearing screening (NHS) and auditory brainstem response (ABR) of the neonatal intensive care unit (NICU) infants.

**Table 1 ijerph-17-08082-t001:** Characteristics of the NICU infants in newborn hearing screening and the diagnosis of hearing loss.

Variables	All Infants (*n* = 2404)	Hearing Loss Group (*n* = 43)	Control Group (*n* = 172)
Gender, male/female (% of male)	1336/1066 (56)	24/19 (55.8)	103/69 (59.9)
Birth weight, kg, median (IQR)	2.61 (2.00, 3.11)	1.23 (0.99, 2.65)	1.20 (0.99, 2.51)
Gestational weeks at birth, median (IQR)	36.0 (33.0, 38.0)	29.0 (27.0, 36.0)	28.5 (27.0, 36.0)
Admission days of NICU, median (IQR)	7.0 (5.0, 21.0)	63.0 (14.0, 88.0)	52.0 (7.0, 77.5)
NHS days after birth, median (IQR)	6.0 (3.0, 18.0)	55.0 (13.0, 77.0)	44.5 (7, 77.5)
Passed infants no. (%)	2227 (92.6)	0 (0)	172 (100)
Referred infants no. (%)	177 (7.4)	43 (100)	0 (0)
Referred on both ears (%)	77 (3.2)	29 (67.4)	0
Referred on left ear only (%)	46 (1.9)	4 (9.3)	0
Referred on right ear only (%)	54 (2.2)	10 (23.3)	0
Diagnostic ABR infants no.	60	43	0
ABR days after birth, median (IQR)	123.0 (59.5, 234.5)	119 (64, 174)	0
ABR rate of referred infants, %	33.9	100	0
Follow-up loss rate, %	66.1	0	0
Hearing loss infants no. (% of total no.)	43 (1.8)	43 (100)	0
Unilateral (% of total no.)	17 (0.7)	17 (36.5)	0
Bilateral (% of total no.)	26 (1.1)	26 (60.5)	0

NICU: neonatal intensive care unit, NHS: newborn hearing screening, ABR: auditory brainstem response, IQR: interquartile range (quartile 1, quartile 3).

**Table 2 ijerph-17-08082-t002:** Patient factors in the control (normal hearing) and hearing loss group in NICU infants.

Variables	Control Group (*n* = 172)	Hearing Loss Group (*n* = 43)	*p*-Value
Gender, male/female (% of male)	103/69 (59.9)	24/19 (55.8)	0.627
No. of each diagnosis (% in group)			
Germinal matrix hemorrhage	13 (7.6)	2 (4.7)	0.741
Respiratory distress syndrome	59 (34.3)	18 (41.9)	0.355
Cerebral bleeding	9 (5.2)	2 (4.7)	1.000
Sepsis	63 (36.6)	27 (62.8)	0.003 *
Pneumonia	13 (7.6)	7 (16.3)	0.078
Facial anomaly	5 (2.9)	2 (4.7)	0.629
Chromosome anomaly	0 (0)	1 (2.3)	0.200
Congenital metabolic disorder	43 (25.0)	11 (25.6)	0.937
Intrauterine infection such as Toxoplasmosis, Rubella, CMV and Herpes virus	3 (1.7)	1 (2.3)	1.000
Apgar score at 1 min, mean (SD)	4.9 (2.7)	4.6 (3.0)	0.469
Apgar score at 5 min, mean (SD)	6.7 (2.4)	6.4 (2.7)	0.373
Peak total bilirubin, mean (SD)	5.1 (4.9)	8.6 (5.8)	0.001 *
Peak direct bilirubin, mean (SD	1.2 (1.6)	1.6 (1.7)	0.112

CMV: cytomegalovirus. * *p* < 0.05 in the Pearson chi-square test and Fisher’s exact test for categorical variables, or the Independent *t*-tests for continuous variables.

**Table 3 ijerph-17-08082-t003:** Treatment factors in the control (normal hearing) and hearing loss group in NICU infants.

Variables	Control Group (*n* = 172)	Hearing Loss Group (*n* = 43)	*p*-Value
Admission days of NICU, median (IQR)	52.0 (7.0, 84.8)	63.0 (14.0, 88.0)	0.443
Ventilation days, median (IQR)	0 (0, 8.0)	1.0 (0, 12.0)	0.693
Phototherapy days, median (IQR)	0 (0, 0)	1.0 (0, 8.0)	0.002*
No. of infants using AG (%)	74 (43.0)	14 (32.6)	0.212
Duration of AG, days, median (IQR)	0 (0, 7.0)	0 (0, 9.0)	0.589
No. of infants using VM (%)	59 (34.3)	19 (44.2)	0.228
Duration of VM, days, median (IQR)	0 (0, 7.0)	0 (0.0, 16.0)	0.024 *
Duration of VM & AG, days, median (IQR)	4.5 (0, 14.0)	0 (7.0, 26.0)	0.111
No. of infants using LIDs (%)	32 (18.6)	17 (39.5)	0.003 *
Duration of LIDs, days, median (IQR)	0 (0, 0)	0 (0, 4.0)	0.880
No. of infants using surfactant (%)	53 (30.8)	18 (41.9)	0.168
No. of infants receiving transfusions (%)	56 (32.6)	25 (58.1)	0.108

NICU: neonatal intensive care unit, AG: aminoglycoside, VM: vancomycin, LIDs: loop-inhibiting diuretics, IQR: interquartile range (quartile 1, quartile 3). * *p* < 0.05 in the Pearson chi-square test and Fisher’s exact test for categorical variables, or the Independent *t*-tests for continuous variables.

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
