# Peer review of "Analysis of the Risk Factors Associated with Hearing Loss of Infants Admitted to a Neonatal Intensive Care Unit: A 13-Year Experience in a University Hospital in Korea"

_ijerph, 2020, doi:10.3390/ijerph17218082_

Round 1

Reviewer 1 Report

This manuscript provides extensive summary data and an analysis of risk factors for hearing loss in a group of 215 babies selected from a larger number who were referred to the NICU. The manuscript is generally well-written, with a detailed Introduction and a Discussion that compares these findings to those of similar analyses from other countries. The findings are broadly consistent with previous reports. There are differences in the details, most likely because in all such studies, the number of cases of hearing loss tends to be low (a good thing, of course). The study does not break much new ground, but does provide archival data that will be of relevance for the field of neonatal hearing loss.

Specific comments:

line 71: A large majority of babies who passed the NHS test were excluded from the control group. The text notes that babies in the control subset were matched for age, sex, and birth year. Were there other criteria? For example, perhaps the first 172 consecutive babies who matched were selected. It would be useful to have more information about the selection process, whatever it was.

line 225: "more than 2000 cases" is misleading at best, when all but 215 cases were excluded from the actual analysis.

Author Response

We thank the reviewers for their valuable time and constructive comments. These comments have helped us to improve the quality of our manuscript. We have reviewed and revised our manuscript in accordance with the reviewers’ recommendations and concerns. Responses and all revisions to the manuscript are shown in Red for the reviewers’ convenience. Thank you so much.

Point 1: line 71: A large majority of babies who passed the NHS test were excluded from the control group. The text notes that babies in the control subset were matched for age, sex, and birth year. Were there other criteria? For example, perhaps the first 172 consecutive babies who matched were selected. It would be useful to have more information about the selection process, whatever it was.

Response 1: Thank you for your kind review and for pointing out my miss for the other criteria. You are right that the first 172 consecutive babies who matched were selected, and I have added this information in the revised manuscript line 75. Thank you again for your advice.

Point 2: line 225: "more than 2000 cases" is misleading at best, when all but 215 cases were excluded from the actual analysis.

Response 2: Your point for this one is exactly correct. So we changed the sentence in the revised manuscript line 233 to “This study is the first one to evaluate the hearing loss in infants in Korea with more than 2,000 cases, and the risk factors of hearing loss with more than 200 cases.” Thank you again for correcting my mistake.

Reviewer 2 Report

The article deals with the subject of hearing loss in patients hospitalized in neonatal intensive care units. The article analyzes and researches statistically significant risk factors for hearing loss in NICU in Korea.

In the materials and methods it would be correct to divide the study participants according to inclusion and exclusion criteria.

In the statistical part, before using the Student T test it is correct to check the normal distribution through the kolmogorov smirnov test.

In the discussion, the bibliography should be supplemented with these articles:

Rizzo S, Bentivegna D, Dispenza F, Mucia M, Plescia F, Thomas E, La Mattina E,, Salvago P, Sireci F, Martines F Audiological Risk Factors and Screening Strategies in NICU Infants. In: Francesco Martines eds. Neonatal Intensive Care Units (NICUs): Clinical and Patient Perspectives, Levels of Care and Emerging Challenges. Chapter 1; Nova publisher

Gazia F, Abita P, Alberti G, Loteta S, Longo P, Caminiti F et al. NICU Infants & SNHL: Experience of a western Sicily tertiary care center. Acta Medica Mediterranea. 2019, 35 (2): 1001-7

Sireci F, Ferrara S, Gargano R, Mucia M, Plescia F, Rizzo S, Salvago P, Martines F Hearing Loss in Neonatal Intensive Care Units (NICUs): Follow-Up Surveillance. In: Francesco Martines eds. Neonatal Intensive Care Units (NICUs): Clinical and Patient Perspectives, Levels of Care and Emerging Challenges. Chapter 2; Nova publisher

In conclusion, the article adds nothing new to the scientific literature, but it is an excellent analysis of the state of the art in Korea regarding the relationship between hearing loss and NICU in Korea.

Author Response

We thank the reviewers for their valuable time and constructive comments. These comments have helped us to improve the quality of our manuscript. We have reviewed and revised our manuscript in accordance with the reviewers’ recommendations and concerns. Responses and all revisions to the manuscript are shown in Red for the reviewers’ convenience. Thank you so much.

Point 1: In the materials and methods it would be correct to divide the study participants according to inclusion and exclusion criteria.

Response 1: First of all, thank you for your kind review. To more accurately describe the inclusion and exclusion criteria of the study, we added “All the NICU infants in the study period with the results of NHS were included. Exclusion criteria included those infants without any of NHS results, and those lacking medical information for any of patient and treatment factors as described below such as infants born in other hospitals and admitted to NICU in the study hospital.” in the revised manuscript line 68 of the materials and methods section as your recommendation.

Point 2: In the statistical part, before using the Student T test it is correct to check the normal distribution through the kolmogorov smirnov test.

Response 2:  We agree with your comment, and we added “Initially, all data were analyzed using Kolmogorov-Smirnov test to assess for normality.” in the revised manuscript line 93. Thank you for pointing out.

Point 3: In the discussion, the bibliography should be supplemented with these articles:

Rizzo S, Bentivegna D, Dispenza F, Mucia M, Plescia F, Thomas E, La Mattina E,, Salvago P, Sireci F, Martines F Audiological Risk Factors and Screening Strategies in NICU Infants. In: Francesco Martines eds. Neonatal Intensive Care Units (NICUs): Clinical and Patient Perspectives, Levels of Care and Emerging Challenges. Chapter 1; Nova publisher

Gazia F, Abita P, Alberti G, Loteta S, Longo P, Caminiti F et al. NICU Infants & SNHL: Experience of a western Sicily tertiary care center. Acta Medica Mediterranea. 2019, 35 (2): 1001-7

Sireci F, Ferrara S, Gargano R, Mucia M, Plescia F, Rizzo S, Salvago P, Martines F Hearing Loss in Neonatal Intensive Care Units (NICUs): Follow-Up Surveillance. In: Francesco Martines eds. Neonatal Intensive Care Units (NICUs): Clinical and Patient Perspectives, Levels of Care and Emerging Challenges. Chapter 2; Nova publisher

Response 3: Thank you for your kind consideration and for recommending good references. We have added all three articles in the References (the references’ number from 12 to 14) and in the revised manuscript line 48 and 213 mentioning the NICU risk factors. Thank you so much.

Reviewer 3 Report

Analysis of the risk factors…

Newborns subjected to neonatal intensive care have a much higher incidence of hearing impairment than corresponding normal babies. In the present manuscript the risk factors of hearing loss in children nurtured at a neonatal intensive care unit (NICU), during a 13 years period, in Seoul, Korea are evaluated and analyzed in a retrospective study. Data were collected for 2404 infants, which had underwent hearing screening tests. 177 infants did not pass these tests of which 43 had a confirmed hearing loss, either bilateral or unilateral. The data for these 43 infants were compared to 172 other NICU infants but without hearing impairment. The presence of sepsis and peak total bilirubin were significantly higher for the hearing loss group compared to the control group. Similarily the days of phototherapy, duration of vancomycin use and the use of loop-inhibiting diuretics differed significantly between the hearing loss group and the controls.

This is the first report to describe the occurrence and risk factors of hearing impairment in infants nurtured at an NICU in Korea. The study also gives insights to variables connected with neonatal hearing screening like methods used and at which times the initial hearing sceening tests are performed. The study is well designed and performed. The results are interesting and convincing. Though a retrospective study I find this manuscript acceptable for publication in Int J Environ Res Public Health in its present form. The authors should however be urged to start a prospective study, in a similar manner, to gain further information on this important area.

Author Response

We thank the reviewers for their valuable time and constructive comments. These comments have helped us to improve the quality of our manuscript. We have reviewed and revised our manuscript in accordance with the reviewers’ recommendations and concerns. Responses and all revisions to the manuscript are shown in Red for the reviewers’ convenience. Thank you so much.

Point 1: The authors should however be urged to start a prospective study, in a similar manner, to gain further information on this important area.

Response 1: Thank you very much for your kind review. As we agree with your comment, we added “Further prospective study is needed ~” in the limitation of the study section in the revised manuscript line 251.

Reviewer 4 Report

This retrospective study is dedicated to the problem of risk factors for hearing loss (HL) in babies received treatment in neonatal intensive care units (NICU). The study is aimed to assess the significance of definite neonatal conditions in the prediction of congenital hearing loss.

The aim of the study is well defined. The design is described in details. The style and structure of the article meet the requirements. The tables and figures are informative and clear. The abstract reflects the content of the Article.

Comments:

Line 68: “Hearing loss was defined as an elevated ABR response threshold (≥35 dB nHL) in the best hearing ear”.

Should be “better hearing ear”. Thereby the protocol refers only bilateral HL. Lower in the Chapter “Classification of groups” (Lines 70-71) it is said “which included unilateral hearing loss”. The target condition is to be clarified.

Line 112 (Figure 1): “Control group (normal hearing)”

It is more correct to define the control group as passed NHS (and this term is used in the text – line 110). The hearing status of these babies wasn’t assessed by full audiological examination.

Lines 120, 140 (Table 1, Table 3): there are many cases presenting quantative data where SD values exceed mean values. This bias could be due to unnormal distribution of data so it could be more correct to present and compare data in median and interquartile range.

References:

Several pairs of different publications probably contain the same content (8 and 12, 11 and 14). Is there a need for duplications?

References 9, 10, 11, 14, 17 – names of organizations are written like personal names (England, P.H. – Public Health England; Society, T.K.A. – The Korean Audiological Society; Organization, W.H. – World Health Organization).

Conclusion: the article could be accepted for publication after minor correction.

Author Response

We thank the reviewers for their valuable time and constructive comments. These comments have helped us to improve the quality of our manuscript. We have reviewed and revised our manuscript in accordance with the reviewers’ recommendations and concerns. Responses and all revisions to the manuscript are shown in Red for the reviewers’ convenience. Thank you so much.

Point 1: Line 68: “Hearing loss was defined as an elevated ABR response threshold (≥35 dB nHL) in the best hearing ear”.

Should be “better hearing ear”. Thereby the protocol refers only bilateral HL. Lower in the Chapter “Classification of groups” (Lines 70-71) it is said “which included unilateral hearing loss”. The target condition is to be clarified.

Response 1: We thank the reviewer for allowing us to clarify this term. As you recommended, we revised the “in the best hearing ear” to “in either ear” to include unilateral hearing loss in the revised manuscript line 68. Thank you so much.

Point 2: Line 112 (Figure 1): “Control group (normal hearing)”

It is more correct to define the control group as passed NHS (and this term is used in the text – line 110). The hearing status of these babies wasn’t assessed by full audiological examination.

Response 2: Thank you again for pointing out our mistake. We have changed “(normal hearing)” to “(Passed NHS)” in Figure 1 as your recommendation.

Point 3: Lines 120, 140 (Table 1, Table 3): there are many cases presenting quantative data where SD values exceed mean values. This bias could be due to unnormal distribution of data so it could be more correct to present and compare data in median and interquartile range.

Response 3: Thank you for your thoughtful comments. As you mentioned, we presented the median and interquartile range (IQR) in Tables 1 and 3 and result section instead of the mean and standard deviation. Thank you so much.

Point 4: References: Several pairs of different publications probably contain the same content (8 and 12, 11 and 14). Is there a need for duplications?

References 9, 10, 11, 14, 17 – names of organizations are written like personal names (England, P.H. – Public Health England; Society, T.K.A. – The Korean Audiological Society; Organization, W.H. – World Health Organization).

Response 4: Thank you for pointing out the duplications. We have erased the duplications and revised the names in References 9, 10, 11, 14, 17.
